# Graphical model inference: Sequential Monte Carlo meets deterministic approximations

**Fredrik Lindsten**
Department of Information Technology
Uppsala University
Uppsala, Sweden
`fredrik.lindsten@it.uu.se`

**Jouni Helske**
Department of Science and Technology
Linköping University
Norrköping, Sweden
`jouni.helske@liu.se`

**Matti Vihola**
Department of Mathematics and Statistics
University of Jyväskylä
Jyväskylä, Finland
`matti.s.vihola@jyu.fi`

## Abstract

Approximate inference in probabilistic graphical models (PGMs) can be grouped into deterministic methods and Monte-Carlo-based methods. The former can often provide accurate and rapid inferences, but are typically associated with biases that are hard to quantify. The latter enjoy asymptotic consistency, but can suffer from high computational costs. In this paper we present a way of bridging the gap between deterministic and stochastic inference. Specifically, we suggest an efficient sequential Monte Carlo (SMC) algorithm for PGMs which can leverage the output from deterministic inference methods. While generally applicable, we show explicitly how this can be done with loopy belief propagation, expectation propagation, and Laplace approximations. The resulting algorithm can be viewed as a post-correction of the biases associated with these methods and, indeed, numerical results show clear improvements over the baseline deterministic methods as well as over "plain" SMC.

## 1  Introduction

Probabilistic graphical models (PGMs) are ubiquitous in machine learning for encoding dependencies in complex and high-dimensional statistical models [18]. Exact inference over these models is intractable in most cases, due to non-Gaussianity and non-linear dependencies between variables. Even for discrete random variables, exact inference is not possible unless the graph has a tree-topology, due to an exponential (in the size of the graph) explosion of the computational cost. This has resulted in the development of many approximate inference methods tailored to PGMs. These methods can roughly speaking be grouped into two categories: *(i)* methods based on deterministic (and often heuristic) approximations, and *(ii)* methods based on Monte Carlo simulations.

The first group includes methods such as Laplace approximations [30], expectation propagation [23], loopy belief propagation [26], and variational inference [36]. These methods are often promoted as being fast and can reach higher accuracy than Monte-Carlo-based methods for a fixed computational cost. The downside, however, is that the approximation errors can be hard to quantify and even if the computational budget allows for it, simply spending more computations to improve the accuracy can be difficult. The second group of methods, including Gibbs sampling [28] and sequential Monte Carlo (SMC) [11, 24], has the benefit of being asymptotically consistent. That is, under mild assumptions

they can often be shown to converge to the correct solution if simply given enough compute time. The problem, of course, is that "enough time" can be prohibitively long in many situations, in particular if the sampling algorithms are not carefully tuned.

In this paper we propose a way of combining deterministic inference methods with SMC for inference in general PGMs expressed as factor graphs. The method is based on a sequence of artificial target distributions for the SMC sampler, constructed via a sequential graph decomposition. This approach has previously been used by [24] for enabling SMC-based inference in PGMs. The proposed method has one important difference however; we introduce a so called twisting function in the targets obtained via the graph decomposition which allows for taking dependencies on "future" variables of the sequence into account. Using twisted target distributions for SMC has recently received significant attention in the statistics community, but to our knowledge, it has mainly been developed for inference in state space models [14, 16, 34, 31]. We extend this idea to SMC-based inference in general PGMs, and we also propose a novel way of constructing the twisting functions, as described below. We show in numerical illustrations that twisting the targets can significantly improve the performance of SMC for graphical models.

A key question when using this approach is how to construct efficient twisting functions. Computing the *optimal twisting functions* boils down to performing exact inference in the model, which is assumed to be intractable. However, this is where the use of deterministic inference algorithms comes into play. We show how it is possible to compute sub-optimal, but nevertheless efficient, twisting functions using some popular methods—Laplace approximations, expectation propagation and loopy belief propagation. Furthermore, the framework can easily be used with other methods as well, to take advantage of new and more efficient methods for approximate inference in PGMs.

The resulting algorithm can be viewed as a post-correction of the biases associated with the deterministic inference method used, by taking advantage of the rigorous convergence theory for SMC (see e.g., [9]). Indeed, the approximation of the twisting functions only affect the efficiency of the SMC sampler, not its asymptotic consistency, nor the unbiasedness of the normalizing constant estimate (which is a key merit of SMC samplers). An implication of the latter point is that the resulting algorithm can be used together with pseudo-marginal [1] or particle Markov chain Monte Carlo (MCMC) [3] samplers, or as a post-correction of approximate MCMC [34]. This opens up the possibility of using well-established approximate inference methods for PGMs in this context.

**Additional related work:** An alternative approach to SMC-based inference in PGMs is to make use of tempering [10]. For discrete models, [15] propose to start with a spanning tree to which edges are gradually added within an SMC sampler to recover the original model. This idea is extended by [6] by defining the intermediate targets based on conditional mean field approximations. Contrary to these methods our approach can handle both continuous and/or non-Gaussian interactions, and does not rely on intermediate MCMC steps within each SMC iteration. When it comes to combining deterministic approximations and Monte-Carlo-based inference, previous work has largely been focused on using the approximation as a proposal distribution for importance sampling [13] or MCMC [8]. Our method has the important difference that we do not only use the deterministic approximation to design the proposal, but also to select the intermediate SMC targets via the design of efficient twisting functions.

## 2 Setting the stage

### 2.1 Problem formulation

Let $\pi(x_{1:T})$ denote a distribution of interest over a collection of random variables $x_{1:T} = \{x_1, \ldots, x_T\}$. The model may also depend on some "top-level" hyperparameters, but for brevity we do not make this dependence explicit. In Bayesian statistics, $\pi$ would typically correspond to a posterior distribution over some latent variables given observed data. We assume that there is some structure in the model which is encoded in a factor graph representation [20],

$$\pi(x_{1:T}) = \frac{1}{Z} \prod_{j \in \mathcal{F}} f_j(x_{\mathcal{I}_j}), \tag{1}$$

where $\mathcal{F}$ denotes the set of factors, $\mathcal{I} := \{1, \ldots, T\}$ is the set of variables, $\mathcal{I}_j$ denotes the index set of variables on which factor $f_j$ depends, and $x_{\mathcal{I}_j} := \{x_t : t \in \mathcal{I}_j\}$. Note that $\mathcal{I}_j = \text{Ne}(j)$ is simply the set of neighbors of factor $f_j$ in the graph (recall that in a factor graph all edges are between factor

nodes and variable nodes). Lastly, $Z$ is the normalization constant, also referred to as the partition function of the model, which is assumed to be intractable. The factor graph is a general representation of a probabilistic graphical model and both directed and undirected PGMs can be written as factor graphs. The task at hand is to approximate the distribution $\pi(x_{1:T})$, as well as the normalizing constant $Z$. The latter plays a key role, e.g., in model comparison and learning of top-level model parameters.

## 2.2 Sequential Monte Carlo

Sequential Monte Carlo (SMC, see, e.g., [11]) is a class of importance-sampling-based algorithms that can be used to approximate some, quite arbitrary, sequence of probability distributions of interest. Let

$$\pi_t(x_{1:t}) = \frac{\gamma_t(x_{1:t})}{Z_t}, \qquad\qquad t = 1, \ldots, T,$$

be a sequence of probability density functions defined on spaces of increasing dimension, where $\gamma_t$ can be evaluated point-wise and $Z_t$ is a normalizing constant. SMC approximates each $\pi_t$ by a collection of $N$ weighted particles $\{(x_{1:t}^i, w_t^i)\}_{i=1}^N$, generated according to Algorithm 1.

---

**Algorithm 1** Sequential Monte Carlo (all steps are for $i = 1, \ldots, N$)

1. Sample $x_1^i \sim q_1(x_1)$, set $\widetilde{w}_1^i = \gamma_1(x_1^i)/q_1(x_1^i)$ and $w_1^i = \widetilde{w}_1^i / \sum_{j=1}^N \widetilde{w}_1^j$.

2. **for** $t = 2, \ldots, T$:

   (a) *Resampling:* Simulate ancestor indices $\{a_t^i\}_{i=1}^N$ with probabilities $\{\nu_{t-1}^i\}_{i=1}^N$.

   (b) *Propagation:* Simulate $x_t^i \sim q_t(x_t|x_{1:t-1}^{a_t^i})$ and set $x_{1:t}^i = \{x_{1:t-1}^{a_t^i}, x_t^i\}$.

   (c) *Weighting:* Compute $\widetilde{w}_t^i = \omega_t(x_{1:t}^i) w_{t-1}^{a_t^i} / \nu_{t-1}^{a_t^i}$ and $w_t^i = \widetilde{w}_t^i / \sum_{j=1}^N \widetilde{w}_t^j$.

---

In step 2(a) we use arbitrary resampling weights $\{\nu_{t-1}^i\}_{i=1}^N$, which may depend on all variables generated up to iteration $t-1$. This allows for the use of look-ahead strategies akin to the auxiliary particle filter [27], as well as adaptive resampling based on *effective sample size* (ESS) [19]: if the ESS is below a given threshold, say $N/2$, set $\nu_{t-1}^i = w_{t-1}^i$ to resample according to the importance weights. Otherwise, set $\nu_{t-1}^i \equiv 1/N$ which, together with the use of a low-variance (e.g., stratified) resampling method, effectively turns the resampling off at iteration $t$.

At step 2(b) the particles are propagated forward by simulating from a user-chosen proposal distribution $q_t(x_t|x_{1:t-1})$, which may depend on the complete history of the particle path. The *locally optimal proposal*, which minimizes the conditional weight variance at iteration $t$, is given by

$$q_t(x_t|x_{1:t-1}) \propto \gamma_t(x_{1:t})/\gamma_{t-1}(x_{1:t-1}) \tag{2}$$

for $t \geq 2$ and $q_1(x_1) \propto \gamma_1(x_1)$. If, in addition to using the locally optimal proposal, the resampling weights are computed as $\nu_{t-1}^i \propto \int \gamma_t(\{x_{1:t-1}^i, x_t\})dx_t/\gamma_{t-1}(x_{1:t-1}^i)$, then the SMC sampler is said to be *fully adapted*. At step 2(c) new importance weights are computed using the weight function $\omega_t(x_{1:t}) = \gamma_t(x_{1:t})/\left(\gamma_{t-1}(x_{1:t-1})q_t(x_t|x_{1:t-1})\right)$.

The weighted particles generated by Algorithm 1 can be used to approximate each $\pi_t$ by the empirical distribution $\sum_{i=1}^N w_t^i \delta_{x_{1:t}^i}(dx_{1:t})$. Furthermore, the algorithm provides unbiased estimates of the normalizing constants $Z_t$, computed as $\widehat{Z}_t = \prod_{s=1}^t \left\{ \frac{1}{N} \sum_{i=1}^N \widetilde{w}_s^i \right\}$; see [9] and the supplementary material.

## 3 Graph decompositions and twisted targets

We now turn our attention to the factor graph (1). To construct a sequence of target distributions for an SMC sampler, [24] proposed to decompose the graphical model into a sequence of sub-graphs, each defining an intermediate target for the SMC sampler. This is done by first ordering the variables, or the factors, of the model in some way—here we assume a fixed order of the variables $x_{1:T}$ as

indicated by the notation; see Section 5 for a discussion about the ordering. We then define a sequence of unnormalized densities $\{\gamma_t(x_{1:t})\}_{t=1}^T$ by gradually including the model variables and the corresponding factors. This is done in such a way that the final density of the sequence includes all factors and coincides with the original target distribution of interest,

$$\gamma_T(x_{1:T}) = \prod_{j \in \mathcal{F}} f_j(x_{\mathcal{I}_j}) \propto \pi(x_{1:T}). \tag{3}$$

We can then target $\{\gamma_t(x_{1:t})\}_{t=1}^T$ with an SMC sampler. At iteration $T$ the resulting particle trajectories can be taken as (weighted) samples from $\pi$, and $\widehat{Z} := \widehat{Z}_T$ will be an unbiased estimate of $Z$.

To define the intermediate densities, let $F_1, \ldots, F_T$ be a partitioning of the factor set $\mathcal{F}$ defined by:

$$F_t = \{j \in \mathcal{F} : t \in \mathcal{I}_j, t+1 \notin \mathcal{I}_j, \ldots, T \notin \mathcal{I}_j\}.$$

In words, $F_t$ is the set of factors depending on $x_t$, and possibly $x_{1:t-1}$, but not $x_{t+1:T}$. Furthermore, let $\mathcal{F}_t = \sqcup_{s=1}^t F_s$. Naesseth et al. [24] defined a sequence of intermediate target densities as[1]

$$\gamma_t(x_{1:t}) = \prod_{j \in \mathcal{F}_t} f_j(x_{\mathcal{I}_j}), \qquad\qquad t = 1, \ldots, T. \tag{4}$$

Since $\mathcal{F}_T = \mathcal{F}$, it follows that the condition (3) is satisfied. However, even though this is a valid choice of target distributions, leading to a consistent SMC algorithm, the resulting sampler can have poor performance. The reason is that the construction (4) neglects the dependence on "future" variables $x_{t+1:T}$ which may have a strong influence on $x_{1:t}$. Neglecting this dependence can result in samples at iteration $t$ which provide an accurate approximation of the intermediate target $\gamma_t$, but which are nevertheless very unlikely under the actual target distribution $\pi$.

To mitigate this issue we propose to use a sequence of *twisted* intermediate target densities,

$$\gamma_t^\psi(x_{1:t}) := \psi_t(x_{1:t})\gamma_t(x_{1:t}) = \psi_t(x_{1:t}) \prod_{j \in \mathcal{F}_t} f_j(x_{\mathcal{I}_j}), \qquad t = 1, \ldots, T-1, \tag{5}$$

where $\psi_t(x_{1:t})$ is an arbitrary positive "twisting function" such that $\int \gamma_t^\psi(x_{1:t})dx_{1:t} < \infty$. (Note that there is no need to explicitly compute this integral as long as it can be shown to be finite.) Twisting functions have previously been used by [14, 16] to "twist" the Markov transition kernel of a state space (or Feynman-Kac) model; we take a slightly different viewpoint and simply consider the twisting function as a multiplicative adjustment of the SMC target distribution.

The definition of the twisted targets in (5) is of course very general and not very useful unless additional guidance is provided. To this end we state the following simple optimality condition (the proof is in the supplementary material; see also [14, Proposition 2]).

**Proposition 1.** *Assume that the twisting functions in* (5) *are given by*

$$\psi_t^*(x_{1:t}) := \int \prod_{j \in \mathcal{F} \setminus \mathcal{F}_t} f_j(x_{\mathcal{I}_j})dx_{t+1:T} \qquad\qquad t = 1, \ldots, T-1, \tag{6}$$

*that the locally optimal proposals* (2) *are used in the SMC sampler, and that* $\nu_t^i = w_t^i$. *Then, Algorithm 1 results in particle trajectories exactly distributed according to* $\pi(x_{1:T})$ *and the estimate of the normalizing constant is exact;* $\widehat{Z} = Z$ *w.p.1.*

Clearly, the optimal twisting functions are intractable in all situations of interest. Indeed, computing (6) essentially boils down to solving the original inference problem. However, guided by this, we will strive to select $\psi_t(x_{1:t}) \approx \psi_t^*(x_{1:t})$. As pointed out above, the approximation error, here, only affects the efficiency of the SMC sampler, not its asymptotic consistency or the unbiasedness of $\widehat{Z}$. Various ways for approximating $\psi_t^*$ are discussed in the next section.

# 4 Twisting functions via deterministic approximations

In this section we show how a few popular deterministic inference methods can be used to approximate the optimal twisting functions in (6), namely loopy belief propagation (Section 4.1), expectation propagation (Section 4.2), and Laplace approximations (Section 4.3). These methods are likely to be useful for computing the twisting functions in many situations, however, we emphasize that they are mainly used to illustrate the general methodology which can be used with other inference procedures as well.

## 4.1 Loopy belief propagation

Belief propagation [26] is an exact inference procedure for tree-structured graphical models, although its "loopy" version has been used extensively as a heuristic approximation for general graph topologies. Belief propagation consists of passing messages:

$$\text{Factor} \rightarrow \text{variable}: \qquad \mu_{j \rightarrow s}(x_s) = \int f_j(x_{\mathcal{I}_j}) \prod_{u \in \text{Ne}(j) \setminus \{s\}} \lambda_{u \rightarrow j}(x_u) dx_{\mathcal{I}_j \setminus \{s\}},$$

$$\text{Variable} \rightarrow \text{factor}: \qquad \lambda_{s \rightarrow j}(x_s) = \prod_{i \in \text{Ne}(s) \setminus \{j\}} \mu_{i \rightarrow s}(x_s).$$

In graphs with loops, the messages are passed until convergence.

To see how loopy belief propagation can be used to approximate the twisting functions for SMC, we start with the following result for tree-structured model (the proof is in the supplementary material).

**Proposition 2.** *Assume that the factor graph with variable nodes $\{1, \ldots, t\}$ and factor nodes $\{f_j : j \in \mathcal{F}_t\}$ form a (connected) tree for all $t = 1, \ldots, T$. Then, the optimal twisting function (6) is given by*

$$\psi_t^*(x_{1:t}) = \prod_{j \in \mathcal{F} \setminus \mathcal{F}_t} \mu_{j \rightarrow (1:t)}(x_{1:t}) \quad where \quad \mu_{j \rightarrow (1:t)}(x_{1:t}) = \prod_{s \in \{1, \ldots, t\} \cap \mathcal{I}_j} \mu_{j \rightarrow s}(x_s). \quad (7)$$

*Remark* 1. The sub-tree condition of Proposition 2 implies that the complete model is a tree, since this is obtained for $t = T$. The connectedness assumption can easily be enforced by gradually growing the tree, lumping model variables together if needed.

While the optimality of (7) only holds for tree-structured models, we can still make use of this expression for models with cycles, analogously to loopy belief propagation. Note that the message $\mu_{j \rightarrow (1:t)}(x_{1:t})$ is the product of factor-to-variable messages going from the non-included factor $j \in \mathcal{F} \setminus \mathcal{F}_t$ to included variables $s \in \{1, \ldots, t\}$. For a tree-based model there is at most one such message (under the connectedness assumption of Proposition 2), whereas for a cyclic model $\mu_{j \rightarrow (1:t)}(x_{1:t})$ might be the product of several "incoming" messages.

It should be noted that the numerous modifications of the loopy belief propagation algorithm that are available can be used within the proposed framework as well. In fact, methods based on tempering of the messages, such as tree-reweighting [35], could prove to be particularly useful. The reason is that these methods counteract the double-counting of information in classical loopy belief propagation, which could be problematic for the following SMC sampler due to an over-concentration of probability mass. That being said, we have found that even the standard loopy belief propagation algorithm can result in efficient twisting, as illustrated numerically in Section 6.1, and we do not pursue message-tempering further in this paper.

## 4.2 Expectation propagation

Expectation propagation (EP, [23]) is based on introducing approximate factors, $\widetilde{f}_j(x_{\mathcal{I}_j}) \approx f_j(x_{\mathcal{I}_j})$ such that

$$\widetilde{\pi}(x_{1:T}) = \frac{\prod_{j \in \mathcal{F}} \widetilde{f}_j(x_{\mathcal{I}_j})}{\int \prod_{j \in \mathcal{F}} \widetilde{f}_j(x_{\mathcal{I}_j}) dx_{1:T}} \quad (8)$$

approximates $\pi(x_{1:T})$, and where the $\widetilde{f}_j$'s are assumed to be simple enough so that the integral in the expression above is tractable. The approximate factors are updated iteratively until some convergence

criterion is met. To update factor $\widetilde{f}_j$, we first remove it from the approximation to obtain the so called *cavity distribution* $\widetilde{\pi}^{-j}(x_{1:T}) \propto \widetilde{\pi}(x_{1:T})/\widetilde{f}_j(x_{\mathcal{I}_j})$. We then compute a new approximate factor $\widetilde{f}_j$, such that $\widetilde{f}_j(x_{\mathcal{I}_j})\widetilde{\pi}^{-j}(x_{1:T})$ approximates $f_j(x_{\mathcal{I}_j})\widetilde{\pi}^{-j}(x_{1:T})$. Typically, this is done by minimizing the Kullback–Leibler divergence between the two distributions. We refer to [23] for additional details on the EP algorithm.

Once the EP approximation has been computed, it can naturally be used to approximate the optimal twisting functions in (6). By simply plugging in $\widetilde{f}_j$ in place of $f_j$ we get

$$\psi_t(x_{1:t}) = \int \prod_{j\in\mathcal{F}\setminus\mathcal{F}_t} \widetilde{f}_j(x_{\mathcal{I}_j})dx_{t+1:T}. \tag{9}$$

Furthermore, the EP approximation can be used to approximate the optimal SMC proposal. Specifically, at iteration $t$ we can select the proposal distribution as

$$q_t(x_t|x_{1:t-1}) = \widetilde{\pi}(x_t|x_{1:t-1}) = \left(\prod_{j\in F_t} \widetilde{f}_j(x_{\mathcal{I}_j})\right) \frac{\int \prod_{j\in\mathcal{F}\setminus\mathcal{F}_t} \widetilde{f}_j(x_{\mathcal{I}_j})dx_{t+1:T}}{\int \prod_{j\in\mathcal{F}\setminus\mathcal{F}_{t-1}} \widetilde{f}_j(x_{\mathcal{I}_j})dx_{t:T}}. \tag{10}$$

This choice has the advantage that the weight function gets a particularly simple form:

$$\omega_t(x_{1:t}) = \frac{\gamma_t^\psi(x_{1:t})}{\gamma_{t-1}^\psi(x_{1:t-1})q_t(x_t|x_{1:t-1})} = \prod_{j\in F_t} \frac{f_j(x_{\mathcal{I}_j})}{\widetilde{f}_j(x_{\mathcal{I}_j})}. \tag{11}$$

### 4.3 Laplace approximations for Gaussian Markov random fields

A specific class of PGMs with a large number of applications in spatial statistics are latent Gaussian Markov random fields (GMRFs, see, e.g., [29, 30]). These models are defined via a Gaussian prior $p(x_{1:T}) = \mathcal{N}(x_{1:T}|\mu, Q^{-1})$ where the precision matrix $Q$ has $Q_{ij} \neq 0$ if and only if variables $x_i$ and $x_j$ share a factor in the graph. When this latent field is combined with some non-Gaussian or non-linear observational densities $p(y_t|x_t)$, $t = 1, \ldots, T$, the posterior $\pi(x_{1:T})$ is typically intractable. However, when $p(y_t|x_t)$ is twice differentiable, it is straightforward to find an approximating Gaussian model based on a Laplace approximation by simple numerical optimization [12, 33, 30], and use the obtained model as a basis of twisted SMC. Specifically, we use

$$\psi_t(x_{1:t}) = \int \prod_{s=t+1}^{T} \left\{\widetilde{p}(y_s|x_s)\right\}p(x_{t+1:T}|x_{1:t})dx_{t+1:T}, \tag{12}$$

where $\widetilde{p}(y_t|x_t) \approx p(y_t|x_t)$, $t = 1, \ldots, T$ are the Gaussian approximations obtained using Laplace's method. For proposal distributions, we simply use the obtained Gaussian densities $\widetilde{p}(x_t|x_{1:t-1}, y_{1:T})$. The weight functions have similar form as in (11), $\omega_t(x_{1:t}) = p(y_t|x_t)/\widetilde{p}(y_t|x_t)$. For state space models, this approach was recently used in [34].

## 5 Practical considerations

A natural question is how to order the variables of the model. In a time series context a trivial processing order exists, but it is more difficult to find an appropriate order for a general PGM. However, in Section 6.3 we show numerically that while the processing order has a big impact on the performance of non-twisted SMC, the effect of the ordering is less severe for twisted SMC. Intuitively this can be explained by the look-ahead effect of the twisting functions: even if the variables are processed in a non-favorable order they will not "come as a surprise".

Still, intuitively a good candidate for the ordering is to make the model as "chain-like" as possible by minimizing the bandwidth (see, e.g., [7]) of the adjacency matrix of the graphical model. A related strategy is to instead minimize the fill-in of the Cholesky decomposition of the full posterior precision matrix. Specifically, this is recommended in the GMRF setting for faster matrix algebra [29] and this is the approach we use in Section 6.3. Alternatively, [25] propose a heuristic method for adaptive order selection that can be used in the context of twisted SMC as well.

Application of twisting often leads to nearly constant SMC weights and good performance. However, the boundedness of the SMC weights is typically not guaranteed. Indeed, the approximations may have lighter tails than the target, which may occasionally lead to very large weights. This is particularly problematic when the method is applied within a pseudo-marginal MCMC scheme, because unbounded likelihood estimators lead to poor mixing MCMC [1, 2]. Fortunately, it is relatively easy to add a 'regularization' to the twisting, which leads to bounded weights. We discuss the regularization in more detail in the supplement.

Finally, we comment on the computational cost of the proposed method. Once a sequence of twisting functions has been found, the cost of running twisted SMC is comparable to that of running non-twisted SMC. Thus, the main computational overhead comes from executing the deterministic inference procedure used for computing the twisting functions. Since the cost of this is independent of the number of particles $N$ used for the subsequent SMC step, the relative computational overhead will diminish as $N$ increases. As for the scaling with problem size $T$, this will very much depend on the choice of deterministic inference procedure, as well as on the connectivity of the graph, as is typical for graphical model inference. It is worth noting, however, that even for a sparse graph the SMC sampler needs to be efficiently implemented to obtain a favorable scaling with $T$. Due to the (in general) non-Markovian dependencies of the random variables $x_{1:T}$, it is necessary to keep track of the complete particle trajectories $\{x_{1:t}^i\}_{i=1}^N$ for each $t = 1, \ldots, T$. Resampling of these trajectories can however result in the copying of large chunks of memory (of the order $Nt$ at iteration $t$), if implemented in a 'straightforward manner'. Fortunately, it is possible to circumvent this issue by an efficient storage of the particle paths, exploiting the fact that the paths tend to coalesce in $\log N$ steps; see [17] for details. We emphasize that this issue is inherent to the SMC framework itself, when applied to non-Markovian models, and does not depend on the proposed twisting method.

# 6    Numerical illustration

We illustrate the proposed twisted SMC method on three PGMs using the three deterministic approximation methods discussed in Section 4. In all examples we compare with the baseline SMC algorithm by [24] and the two samplers are denoted as SMC-Twist and SMC-Base, respectively. While the methods can be used to estimate both the normalizing constant $Z$ and expectations with respect to $\pi$, we focus the empirical evaluation on the former. The reasons for this are: *(i)* estimating $Z$ is of significant importance on its own, e.g., for model comparison and for pseudo-marginal MCMC, *(ii)* in our experience, the accuracy of the normalizing constant estimate is a good indicator for the accuracy of other estimates as well, and *(iii)* the fact that SMC produces unbiased estimates of $Z$ means that we can more easily assess the quality of the estimates. Specifically, $\log \widehat{Z}$—which is what we actually compute—is negatively biased and it therefore typically holds that higher estimates are better.

## 6.1    Ising model

As a first proof of concept we consider a $16 \times 16$ square lattice Ising model with periodic boundary condition,

$$\pi(x_{1:T}) = \frac{1}{Z} \exp \left( \sum_{(i,j) \in \mathcal{E}} J_{ij} x_i x_j + \sum_{i \in \mathcal{I}} H_i x_i \right).$$

where $T = 256$ and $x_i \in \{-1, +1\}$. We let the interactions be $J_{ij} \equiv 0.44$ and the external magnetic field is simulated according to $H_i \overset{\text{i.i.d.}}{\sim} \text{Uniform}(-1, 1)$.

We use the *Left-to-Right* sequential decomposition considered by [24]. For SMC-Twist we use loopy belief propagation to compute the twisting potentials, as described in Section 4.1. Both SMC-Base and SMC-Twist use *fully adapted* proposals, which is possible due to the discrete nature of the problem. Apart for the computational overhead of running the belief propagation algorithm (which is quite small, and independent of the number of particles used in the subsequent SMC algorithm), the computational costs of the two SMC samplers is more or less the same.

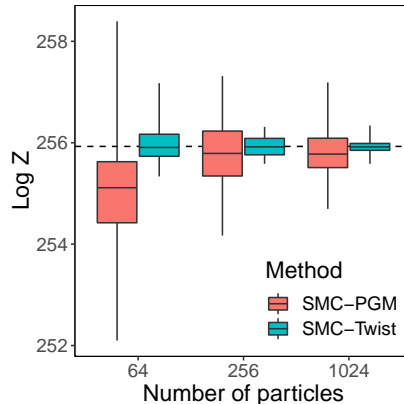

Figure 1:   Results for the Ising model. See text for details.

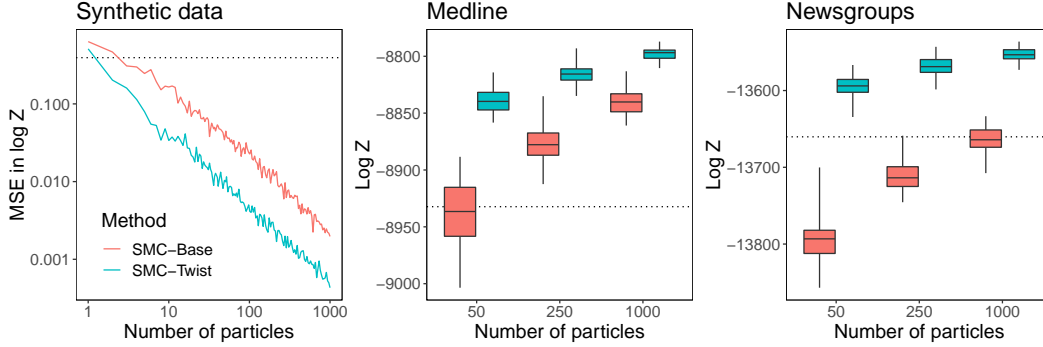

Figure 2: Results for LDA likelihood evaluation for the toy model *(left)*, PubMed data *(mid)*, and 20 newsgroups data *(right)*. Dotted lines correspond to the plain EP estimates. See text for details.

Each algorithm is run 50 times for varying number of particles. Box-plots over the obtained normalizing constant estimates are shown in Figure 1, together with a "ground truth" estimate (dashed line) obtained with an annealed SMC sampler [10] with a large number of particles and temperatures. As is evident from the figure, the twisted SMC sampler outperforms the baseline SMC. Indeed, with twisting we get similar accuracy using $N = 64$ particles, as the baseline SMC with $N = 1024$ particles.

## 6.2 Topic model evaluation

Topic models, such as latent Dirichlet allocation (LDA) [4], are widely used for information retrieval from large document collections. To assess the quality of a learned model it is common to evaluate the likelihood of a set of held out documents. However, this turns out to be a challenging inference problem on its own which has attracted significant attention [37, 5, 32, 22]. Naesseth et al. [24] obtained good performance for this problem with a (non-twisted) SMC method, outperforming the special purpose Left-Right-Sequential sampler by [5]. Here we repeat this experiment and compare this baseline SMC with a twisted SMC. For computing the twisting functions we use the EP algorithm by Minka and Lafferty [22], specifically developed for inference in the LDA model. See [37, 22] and the supplementary material for additional details on the model and implementation details.

First we consider a synthetic toy model with 4 topics and 10 words, for which the exact likelihood can be computed. Figure 2 (left) shows the mean-squared errors in the estimates of the log-likelihood estimates for the two SMC samplers as we increase the number of particles. As can be seen, twisting reduces the error by about half an order-of-magnitude compared to the baseline SMC. In the middle and right panels of Figure 2 we show results for two real datasets, PubMed Central abstracts and 20 newsgroups, respectively (see [37]). For each dataset we compute the log-likelihood of 10 held-out documents. The box-plots are for 50 independent runs of each algorithm, for different number of particles. As pointed out above, due to the unbiasedness of the SMC likelihood estimates it is typically the case that "higher is better". This is also supported by the fact that the estimates increase on average as we increase the number of particles. With this in mind, we see that EP-based twisting significantly improves the performance of the SMC algorithm. Furthermore, even with as few as 50 particles, SMC-Twist clearly improves the results of the EP algorithm itself, showing that twisted SMC can successfully correct for the bias of the EP method.

## 6.3 Conditional autoregressive model with Binomial observations

Consider a latent GMRF $x_{1:T} \sim N(0, \tau Q^{-1})$, where $Q_{tt} = n_t + d$, $Q_{tt'} = -1$ if $t \sim t'$, and $Q_{tt'} = 0$ otherwise. Here $n_t$ is the number of neighbors of $x_t$, $\tau = 0.1$ is a scaling parameter, and $d = 1$ is a regularization parameter ensuring a positive definite precision matrix. Given the latent field we assume binomial observations $y_t \sim \text{Binomial}(10, \text{logit}^{-1}(x_t))$. The spatial structure of the GMRF corresponds to the map of Germany obtained from the R package INLA [21], containing $T = 544$ regions. We simulated one realization of $x_{1:T}$ and $y_{1:T}$ from this configuration and then estimated the log-likelihood of the model 10 000 times with a baseline SMC using a bootstrap proposal, as well as with twisted SMC where the twisting functions were computed using a Laplace

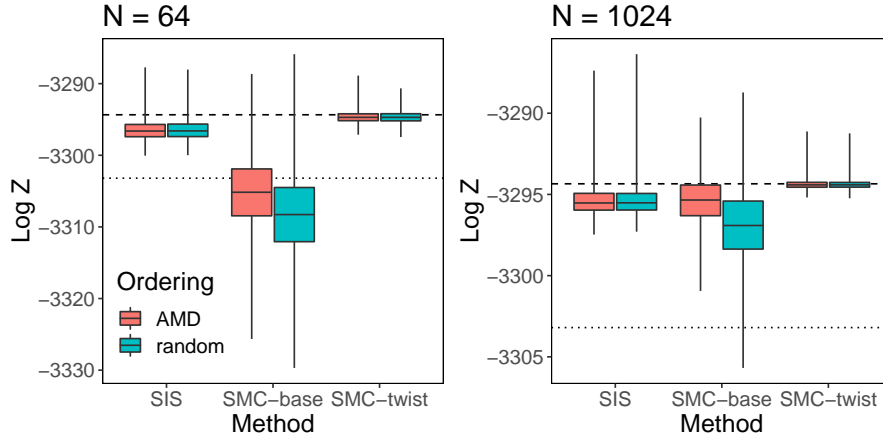

Figure 3: Results for GMRF likelihood evaluation. See text for details.

approximation (see details in the supplementary material). To test the sensitivity of the algorithms to the ordering of the latent variables, we randomly permuted the variables for each replication. We compare this random order with approximate minimum degree reordering (AMD) of the variables, applied before running the SMC. We also varied $N$, the number of particles, from 64 up to 1024. For both SMC approaches, we used adaptive resampling based on effective sample size with threshold of $N/2$. In addition, we ran a twisted sequential importance sampler (SIS), i.e., we set the resampling threshold to zero.

Figure 3 shows the log-likelihood estimates for SMC-Base, SIS and SMC-Twist with $N = 64$ and $N = 1024$ particles, with dashed lines corresponding to the estimates obtained from a single SMC-Twist run with $100\,000$ particles, and dotted lines to the estimates from the Laplace approximation. SMC-Base is highly affected by the ordering of the variables, while the effect is minimal in case of SIS and SMC-Twist. Twisted SMC is relatively accurate already with 64 particles, whereas sequential importance sampling and SMC-Base exhibit large variation and bias still with 1024 particles.

## 7 Conclusions

The twisted SMC method for PGMs presented in this paper is a promising way to combine deterministic approximations with efficient Monte Carlo inference. We have demonstrated how three well-established methods can be used to approximate the optimal twisting functions, but we stress that the general methodology is applicable also with other methods.

An important feature of our approach is that it may be used as 'plug-in' module with pseudo-marginal [1] or particle MCMC [3] methods, allowing for consistent hyperparameter inference. It may also be used as (parallelizable) post-processing of approximate hyperparameter MCMC, which is based purely on deterministic PGM inferences [cf. 34].

An interesting direction for future work is to investigate which properties of the approximations that are most favorable to the SMC sampler. Indeed, it is not necessarily the case that the twisting functions obtained directly from the most accurate deterministic method result in the most efficient SMC sampler. It is also interesting to consider iterative refinements of the twisting functions, akin to the method proposed by [14], in combination with the approach taken here.

### Acknowledgments

FL has received support from the Swedish Foundation for Strategic Research (SSF) via the project *Probabilistic Modeling and Inference for Machine Learning* (contract number: ICA16-0015) and from the Swedish Research Council (VR) via the projects *Learning of Large-Scale Probabilistic Dynamical Models* (contract number: 2016-04278) and *NewLEADS – New Directions in Learning Dynamical Systems* (contract number: 621-2016-06079). JH and MV have received support from the Academy of Finland (grants 274740, 284513 and 312605).

## Footnotes

[1]More precisely, [24] use a fixed ordering of the factors (and not the variables) of the model. They then include one or more additional factors, together with the variables on which these factors depend, in each step of the SMC algorithm. This approach is more or less equivalent to the one adopted here.

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
