[Supplementary Material]

# Supplementary material for 'Graphical model inference: Sequential Monte Carlo meets deterministic approximations'

**Fredrik Lindsten**
Department of Information Technology
Uppsala University
Uppsala, Sweden
`fredrik.lindsten@it.uu.se`

**Jouni Helske**
Department of Science and Technology
Linköping University
Norrköping, Sweden
`jouni.helske@liu.se`

**Matti Vihola**
Department of Mathematics and Statistics
University of Jyväskylä
Jyväskylä, Finland
`matti.s.vihola@jyu.fi`

## Abstract

This is supplementary material for the paper 'Graphical model inference: Sequential Monte Carlo meets deterministic approximations'. We give proofs of two propositions and provide additional details about regularized twisting, the Gaussian Markov random field application, and the topic model evaluation. We also provide a self-contained proof of the unbiasedness of the normalizing constant estimate.

## 1 Proofs

### 1.1 Proof of Proposition 1

Selecting $\psi_t(x_{1:t}) = \psi_t^*(x_{1:t}) = \int \prod_{j \in \mathcal{F} \setminus \mathcal{F}_t} f_j(x_{\mathcal{I}_j}) dx_{t+1:T}$ results in

$$\gamma_t^\psi(x_{1:t}) = \prod_{j \in \mathcal{F}_t} f_j(x_{\mathcal{I}_j}) \int \prod_{j \in \mathcal{F} \setminus \mathcal{F}_t} f_j(x_{\mathcal{I}_j}) dx_{t+1:T}.$$

Consequently,

$$\frac{\gamma_t^\psi(x_{1:t})}{\gamma_{t-1}^\psi(x_{1:t-1})} = \prod_{j \in F_t} f_j(x_{\mathcal{I}_j}) \frac{\int \prod_{j \in \mathcal{F} \setminus \mathcal{F}_t} f_j(x_{\mathcal{I}_j}) dx_{t+1:T}}{\int \prod_{j \in \mathcal{F} \setminus \mathcal{F}_{t-1}} f_j(x_{\mathcal{I}_j}) dx_{t:T}}.$$

This expression integrates to 1, and the locally optimal proposal (given by Eq. (2) in the main paper) is therefore given by $q_t(x_t \mid x_{1:t-1}) = \gamma_t^\psi(x_{1:t})/\gamma_{t-1}^\psi(x_{1:t-1})$ for $t \geq 2$. For $t = 1$ we get $q_1(x_1) \propto \gamma_1^\psi(x_1) = \int \prod_{j \in \mathcal{F}} f_j(x_{\mathcal{I}_j}) dx_{2:T}$, which implies $q_1(x_1) = Z^{-1} \gamma_1^\psi(x_1)$.

We thus get $\widetilde{w}_1^i \equiv Z$ and $\omega_t(x_{1:t}) \equiv 1$ for $t \geq 2$ and thus $\widehat{Z}_T = Z$. Furthermore, this implies that all normalized weights are $\frac{1}{N}$ and the resampling step will therefore not alter the marginal distributions of the particle trajectories. The final particle trajectories are therefore distributed according to

$$q_1(x_1) \prod_{t=2}^T q_t(x_t \mid x_{t-1}) = \frac{\gamma_1^\psi(x_1)}{Z} \prod_{t=2}^T \frac{\gamma_t^\psi(x_{1:t})}{\gamma_{t-1}^\psi(x_{1:t-1})} = \pi(x_{1:T}).$$

In fact, since all importance weights are equal there is no need for resampling. Equivalently, if we use a low-variance resampling method (such as stratified or systematic) then the resampling step will output exactly one copy of each particle, and the resampling if effectively turned off. This implies that the $N$ final trajectories $\{x_{1:T}^i\}_{i=1}^N$ are i.i.d. draws from $\pi$.

## 1.2 Proof of Proposition 2

For a tree-structured factor graph, let $\mathcal{F}_j^{\backslash s}$ denote the set of factors in the subtree, containing factor $f_j$, obtained by removing the edge between factor $f_j$ and variable $x_s$. Furthermore, let $X_j^{\backslash s}$ denote all the variables contained in this subtree. It then holds (see, e.g., [1]) that

$$\mu_{j\to s}(x_s) = \int \prod_{i\in\mathcal{F}_j^{\backslash s}} f_i(x_{I_i})dX_j^{\backslash s}. \tag{1}$$

Now, let $t$ be a fixed iteration index. By assumption the sub-graph with variable nodes $\{1, \ldots, t\}$ and factor nodes $\{f_j : j \in \mathcal{F}_t\}$ is a tree. Since the complete model is also assumed to be a tree, this implies that any factor $j \in \mathcal{F} \setminus \mathcal{F}_t$ is connected to at most one variable node in $\{1, \ldots, t\}$. Specifically, let $\mathcal{J} \subset \mathcal{F} \setminus \mathcal{F}_t$ denote the set of factors such that there exists an edge between $j \in \mathcal{J}$ and some variable $s_j \in \{1, \ldots, t\}$.

It then follows that the the optimal twisting function (Eq. (6) in the main paper) can be factorized as

$$\psi_t^*(x_{1:t}) = \int \prod_{j\in\mathcal{F}\setminus\mathcal{F}_t} f_j(x_{\mathcal{I}_j})dx_{t+1:T}$$

$$= \prod_{j\in\mathcal{J}} \int \prod_{i\in\mathcal{F}_j^{\backslash s_j}} f_i(x_{I_i})dX_j^{\backslash s_j}$$

$$= \prod_{j\in\mathcal{J}} \mu_{j\to s_j}(x_{s_j}).$$

However, by the definition of $\mu_{j\to(1:t)}(x_{1:t})$ it also holds, for a tree-structured graph, that

$$\mu_{j\to(1:t)}(x_{1:t}) = \begin{cases} \mu_{j\to s_j}(x_{s_j}) & \text{if } j \in \mathcal{J}, \\ 1 & \text{otherwise}, \end{cases}$$

which completes the proof.

## 2 Implementation details for topic model evaluation

In this section we present additional details on the model and implementation used in Section 6.2 of the main paper. Matlab code is available on GitHub[1].

The LDA model is given by

$$\pi(\theta, x_{1:T}) \propto \text{Dir}(\theta \mid \alpha) \prod_{t=1}^T \theta_{x_t}\Phi_{w_t x_t} \tag{2}$$

where $\text{Dir}(\theta \mid \alpha)$ is a $K$-dimensional Dirichlet prior over the topic distribution $\theta$. The words of the document, $w_1, \ldots, w_T$, are encoded as integers in $\{1, \ldots, V\}$, where $V$ is the size of the vocabulary. The variable $x_t \in \{1, \ldots, K\}$ is the (latent) topic of word $w_t$, and $\Phi_{:,k}$ is the probability vector over words for topic $k$. For model evaluation we assume that the word distributions $\Phi$ and the concentration parameter for the topic distribution prior $\alpha$ are known (pre-learned), whereas the topic distribution vector $\theta$ as well as the topics $x_{1:T}$ are latent. See [11] for additional details on the model.

The task is to compute the normalizing constant of (2). To this end, Minka and Lafferty [6] proposed an EP algorithm which works as follows. First we marginalize the latent topics,

$$\pi(\theta) \propto \text{Dir}(\theta \mid \alpha) \prod_{t=1}^T \left\{ \sum_{k=1}^K \theta_k \Phi_{w_t k} \right\} = \text{Dir}(\theta \mid \alpha) \prod_{w=1}^V \left\{ \sum_{k=1}^K \theta_k \Phi_{wk} \right\}^{n_w}, \tag{3}$$

where $n_w$ is the number of occurrences of word $w$ in the document. Next, we introduce approximate factors

$$\sum_{k=1}^{K} \theta_k \Phi_{wk} \approx s_w \prod_{k=1}^{K} \theta_k^{\beta_{wk}}, \tag{4}$$

where the $s_w$'s and $\beta_{wk}$'s are updated one word at a time by moment matching, until convergence. These updates are not guaranteed to result in a proper approximate distribution, so therefore Minka and Lafferty [6] propose to skip any update that results in an improper approximation and simply continue with the next word.

For the twisted SMC algorithm we obtain the following expression for the optimal twisting functions:

$$\psi_t^*(\theta, x_{1:t}) = \psi_t^*(\theta) = \sum_{x_{t+1:T}} \prod_{s=t+1}^{T} \theta_{x_s} \Phi_{w_s x_s} = \prod_{s=t+1}^{T} \left\{ \sum_{k=1}^{K} \theta_k \Phi_{w_s k} \right\}. \tag{5}$$

Thus, we can naturally use the EP approximation (4) to define

$$\psi_t(\theta) = \prod_{s=t+1}^{T} \prod_{k=1}^{K} \theta_k^{\beta_{w_s k}}. \tag{6}$$

Combining this with the non-twisted (unnormalized) target $\gamma_t(\theta, x_{1:t}) = \mathrm{Dir}(\theta \mid \alpha) \prod_{s=1}^{t} \theta_{x_s} \Phi_{w_s x_s}$ we get the twisted target $\gamma_t^{\psi}(\theta, x_{1:t}) = \mathrm{Dir}(\theta \mid g_t) \prod_{s=1}^{t} \theta_{x_s} \Phi_{w_s x_s}$ where $g_t = \alpha + \sum_{s=1}^{t} \beta_{w_s,:}$. To ensure proper intermediate targets for the SMC sampler we extend the safety-check mentioned above, and only apply an EP update if all resulting $g_t$'s are positive. We have found that running the EP updates in reverse order, from $t = T$ to $t = 1$, resulted in few skipped updates.

Finally, similarly to [7] we run a Rao-Blackwellized SMC sampler and analytically marginalize $\theta$ conditionally on $x_{1:t}$ for each particle $\{x_{1:t}^i\}_{i=1}^{N}$.

## 3 Implementation details for latent Gaussian Markov field evaluation

In this section we present additional details on the latent Gaussian Markov random field (GMRF) model of Section 4.3 of the main paper. An R package for obtaining the results of Section 6.3 is also available on GitHub[2].

Let $y_t | x_1, \ldots, x_T \sim p(y_t | x_t)$, where $x = (x_1, \ldots, x_T)^{\mathsf{T}}$ is a GMRF with prior mean vector $\mu$ and precision matrix $Q$. For simplicity, we assume that each $y_t$ and $x_t$ is univariate, and that $p(y_t | x_t)$ belongs to the exponential family (see [9] for more general treatment of obtaining Laplace approximations for latent GMRF models). Then

$$p(x|y) \propto \exp \left( -\frac{1}{2}(x - \mu)^{\mathsf{T}} Q (x - \mu) + \sum_{t=1}^{T} \log p(y_t | x_t) \right).$$

Now we use second-order Taylor approximation of $\sum_{t=1}^{T} \log p(y_t | x_t)$ around $\widetilde{x}$. Denote $\dot{\xi}(\widetilde{x}_t)$ as the value of the first derivative of $\log p(y_t | z)$ w.r.t. $z$ at $z = \widetilde{x}_t$, and similarly $\ddot{\xi}(\widetilde{x}_t)$ for the second derivative. Then

$$\widetilde{p}(x|y) \propto \exp \left( -\frac{1}{2} x^{\mathsf{T}} Q x + \mu^{\mathsf{T}} Q x + \sum_{t=1}^{T} (a_t + b_t x_t - \frac{1}{2} c_t x_t^2) \right)$$

$$\propto \exp \left( -\frac{1}{2} x^{\mathsf{T}} (Q + \mathrm{diag}(c)) x + (Q\mu + b)^{\mathsf{T}} x \right),$$

[2]https://github.com/helske/particlefield

where

$$a_t = \log p(y_t|\widetilde{x}_t) - b_t\widetilde{x}_t + \frac{1}{2}c_t\widetilde{x}_t^2,$$
$$b_t = \dot{\xi}(\widetilde{x}_t) + c_t\widetilde{x}_t,$$
$$c_t = -\ddot{\xi}(\widetilde{x}_t).$$

Now given our guess $\widetilde{x}$, we have a Gaussian approximation of the posterior density of $x$, given as a canonical parametrization $\mathcal{N}_c(Q\mu + b, Q + \text{diag}(c))$. Next we can expand again using the point $Q\mu + b$, and repeat until convergence. This gives as an approximating Gaussian model with posterior precision matrix $\widetilde{Q} = Q + \text{diag}(\widehat{c})$ and mean vector $\widetilde{\mu} = Q\mu + \widehat{b}$, with the same posterior mode $\widehat{x}$ as our original model. We follow [10, 4] and use $\widetilde{Z}p(y|\widehat{x})/\widetilde{p}(y|\widehat{x})$ as an approximate likelihood, where $\widetilde{Z}$ is the likelihood of the approximating model, and the ratio term is an approximation of $\mathbb{E}[p(y|x)/\widetilde{p}(y|x)]$ .

For twisted SMC we define the observational level densities as $\widetilde{p}(y_t|x_t) = \exp(\widehat{a}_t + \widehat{b}_t x_t - \frac{1}{2}\widehat{c}_t x_t^2)$, and sample from $\widetilde{p}(x_t|x_{1:t-1}, y_{1:T})$. In order to sample from from this distribution, we will first order $x$ from last to first for easier bookkeeping, i.e. we write

$$\widetilde{p}(x_{1:T}|y_{1:T}) = \mathcal{N}\left(\begin{bmatrix} x_T \\ \vdots \\ x_1 \end{bmatrix} \middle| \begin{bmatrix} \widetilde{\mu}_T \\ \vdots \\ \widetilde{\mu}_1 \end{bmatrix}, \{L_T L_T^\mathsf{T}\}^{-1}\right),$$

where $L_T$ is a Cholesky factor for $\widetilde{Q}$. Denote also the lower right $t \times t$ block of $L_T$ as

$$L_t = \begin{bmatrix} \widehat{L}_t & 0 \\ \widehat{L}_{t-1,t} & L_{t-1} \end{bmatrix}.$$

Now by marginalization and conditioning on $x_{1:t-1}$ we have

$$\widetilde{p}(x_t \mid x_{1:t-1}, y_{1:T}) = \mathcal{N}(x_t \mid \widetilde{\mu}_{t|t-1}, \{\widehat{L}_t\widehat{L}_t^\mathsf{T}\}^{-1}),$$

with

$$\widetilde{\mu}_{t|t-1} = \widetilde{\mu}_t - \frac{\widehat{L}_{t-1,t}^\mathsf{T}}{\widehat{L}_t}\left(\begin{bmatrix} x_{t-1} \\ \vdots \\ x_1 \end{bmatrix} - \begin{bmatrix} \widetilde{\mu}_{t-1} \\ \vdots \\ \widetilde{\mu}_1 \end{bmatrix}\right).$$

## 4 Modified twisting functions which ensure bounded SMC weights

As noted in [5], a direct approximation of the optimal twisting function may lead to unbounded SMC weights, which may cause unstable behavior. This can often be resolved by a regularization. We review how such regularization can be applied in the setting of expectation propagation; the application in GMRF and LBP follow similar steps. Suppose that we have an approximately optimal twisting function of the form

$$\psi_t(x_{1:t}) := \int \prod_{j \in \mathcal{F} \setminus \mathcal{F}_t} \widetilde{f}_j(x_{\mathcal{I}_j}) dx_{t+1:T},$$

where $\widetilde{f}_j$ form our approximate model. Now, let $\widetilde{\psi}_t(x_{1:t}) := \psi_t(x_{1:t}) + \epsilon$, with $\epsilon \geq 0$ being a constant 'regularization' factor. Let $\gamma_t(x_{1:t}) := \prod_{j \in \mathcal{F}_t} f_j(x_{\mathcal{I}_j})$ denote the 'untwisted' unnormalized targets and $\gamma_t^{\widetilde{\psi}} := \gamma_t\widetilde{\psi}_t$ the 'twisted' unnormalized targets. We may now use a proposal of the form

$$q_t(x_t \mid x_{1:t-1}) = \left[1 - \lambda_t(x_{1:t-1})\right]\widetilde{\pi}_t(x_t \mid x_{1:t-1}) + \lambda_t(x_{1:t-1})s_t(x_t \mid x_{1:t-1}),$$

where $s_t(x_t \mid x_{1:t-1})$ is a 'safeguard proposal',

$$\widetilde{\pi}_t(x_t \mid x_{1:t-1}) := \left(\prod_{j \in F_t} \widetilde{f}_j(x_{\mathcal{I}_j})\right)\frac{\psi_t(x_{1:t})}{\psi_{t-1}(x_{1:t-1})}$$

is the 'approximately optimal' proposal, and the mixture weights $\lambda_t(x_{1:t-1}) \in [0, 1]$ are defined as

$$\lambda_t(x_{1:t-1}) := \frac{\epsilon}{\psi_{t-1}(x_{1:t-1}) + \epsilon}.$$

The SMC weights now take the form

$$
\begin{aligned}
\omega_t(x_{1:t}) &= \frac{\gamma_t^{\widetilde{\psi}}(x_{1:t})}{\gamma_{t-1}^{\widetilde{\psi}}(x_{1:t-1})q_t(x_t \mid x_{1:t-1})} \\
&= \frac{\gamma_t(x_{1:t})\big[\psi_t(x_{1:t}) + \epsilon\big]}{\gamma_{t-1}(x_{1:t-1})\big[\prod_{j \in F_t} \widetilde{f}_j(x_{\mathcal{I}_j})\psi_t(x_{1:t}) + \epsilon s_t(x_t \mid x_{1:t-1})\big]} \\
&= \frac{\prod_{j \in F_t} f_j(x_{\mathcal{I}_j})\big[\psi_t(x_{1:t}) + \epsilon\big]}{\prod_{j \in F_t} \widetilde{f}_j(x_{\mathcal{I}_j})\psi_t(x_{1:t}) + \epsilon s_t(x_t \mid x_{1:t-1})}.
\end{aligned}
$$

Note that if $\epsilon = 0$, this reduces to the simple form stated in the main paper. But if $\epsilon > 0$, $\psi_t$ are bounded, and $s_t$ are 'safe' SMC proposals for the untwisted model, that is, $s_t(x_t \mid x_{1:t-1}) \geq \delta \prod_{j \in F_t} f_j(x_{\mathcal{I}_j})$ for some $\delta > 0$, then the SMC weights $\omega_t$ are bounded.

## 5 Unbiasedness of the normalizing constant estimate

It is well known that the SMC normalizing constant estimate is unbiased, i.e., $\mathbb{E}[\widehat{Z}_t] = Z_t$; see, e.g., [2, 12, 8, 7]. However, there are many (equivalent) formulations of generic SMC algorithms presented in the literature, and therefore also many (equivalent) expressions for the normalizing constant estimate. For instance, the estimator is sometimes explicitly modified to take ESS-based resampling into account [3, 12], and sometimes it is expressed in terms of so called adjustment multiplier weights [7]. However, the simple form of the normalizing constant estimator

$$\widehat{Z}_t = \prod_{s=1}^{t} \left\{ \frac{1}{N} \sum_{i=1}^{N} \widetilde{w}_s^i \right\} \tag{7}$$

is in fact valid for any instance of Algorithm 1 (see the main paper), *as long as the unnormalized weights $\widetilde{w}_t^i$ are computed as stated in the algorithm*:

$$\widetilde{w}_t^i = \omega_t(x_{1:t}^i)w_{t-1}^{a_t^i}/\nu_{t-1}^{a_t^i}$$

with $\omega_t(x_{1:t}) = \gamma_t(x_{1:t})/\left(\gamma_{t-1}(x_{1:t-1})q_t(x_t \mid x_{1:t-1})\right)$, for $t \geq 2$, $\widetilde{w}_1^i = \gamma_1(x_1^i)/q_1(x_1^i)$ and $w_t^i = \widetilde{w}_t^i/\sum_{j=1}^{N} \widetilde{w}_t^j$. In particular, as argued in the main paper this includes ESS-based resampling: set the resampling probabilities $\nu_{t-1}^i \equiv 1/N$ and use a low-variance resampling method whenever the ESS is above the resampling threshold, which effectively turns the resampling off.[3] We can thus use the simple expression (7) also in such situations.

For completeness we therefore provide a proof of the unbiasedness of (7) (for any instance of Algorithm 1 of the main paper) below. The proof itself is not new and closely follows [8, 7].

Let $\mathcal{G}_t = \sigma\left(\{x_1^i\}_{i=1}^N, \{x_s^i, a_s^i\}_{i=1}^N : s = 2, \ldots, t\right)$ denote the filtration generated by all random variables simulated in Algorithm 1 up until iteration $t$. We assume that the resampling probabilities $\{\nu_{t-1}^i\}_{i=1}^N$ used at iteration $t$ are $\mathcal{G}_{t-1}$-measureable and that the resampling method is unbiased:

$$\mathbb{E}\left[\sum_{i=1}^{N} \mathbb{1}(a_t^i = j) \,\middle|\, \mathcal{G}_{t-1}\right] = N\nu_{t-1}^j, \qquad j = 1, \ldots, N. \tag{8}$$

Let $t$ be a fixed index and define recursively the functions $f_t(x_{1:t}) \equiv 1$ and

$$f_s(x_{1:s}) = \frac{\int f_{s+1}(x_{1:s+1})\gamma_{s+1}(x_{1:s+1})dx_{s+1}}{\gamma_s(x_{1:s})}$$

for $s = t-1$, $t-2$, ..., 1. Let

$$Q_s = \left( \frac{1}{N} \sum_{i=1}^{N} \widetilde{w}_s^i f_s(x_{1:s}^i) \right) \prod_{u=1}^{s-1} \left\{ \frac{1}{N} \sum_{i=1}^{N} \widetilde{w}_u^i \right\}$$

for $s = 1$, ..., $t$. Note that $Q_t = \widehat{Z}_t$.

Now, for $2 \leq s \leq t$, consider

$$\mathbb{E}[Q_s \mid \mathcal{G}_{s-1}] = \mathbb{E}\left[ \frac{1}{N} \sum_{i=1}^{N} \widetilde{w}_s^i f_s(x_{1:s}^i) \;\middle|\; \mathcal{G}_{s-1} \right] \times \prod_{u=1}^{s-1} \left\{ \frac{1}{N} \sum_{i=1}^{N} \widetilde{w}_u^i \right\}$$

where the first factor of the right-hand-side can be written as

$$\mathbb{E}\left[ \frac{1}{N} \sum_{i=1}^{N} \frac{w_{s-1}^{a_s^i}}{\nu_{s-1}^{a_s^i}} \int \omega_s((x_{1:s-1}^{a_s^i}, x_s)) f_s((x_{1:s-1}^{a_s^i}, x_s)) q(x_s \mid x_{1:s-1}^{a_s^i}) dx_s \;\middle|\; \mathcal{G}_{s-1} \right]$$

$$= \sum_{j=1}^{N} \frac{w_{s-1}^j}{\nu_{s-1}^j} f_{s-1}(x_{1:s-1}^j) \times \mathbb{E}\left[ \frac{1}{N} \sum_{i=1}^{N} \mathbb{1}(a_s^i = j) \;\middle|\; \mathcal{G}_{s-1} \right] = \sum_{j=1}^{N} w_{s-1}^j f_{s-1}(x_{1:s-1}^j)$$

and where we have used (8) for the last equality. It follows that

$$\mathbb{E}[Q_s \mid \mathcal{G}_{s-1}] = \sum_{i=1}^{N} \frac{\widetilde{w}_{s-1}^i}{\sum_{j=1}^{N} \widetilde{w}_{s-1}^j} f_{s-1}(x_{1:s-1}^i) \times \prod_{u=1}^{s-1} \left\{ \frac{1}{N} \sum_{i=1}^{N} \widetilde{w}_u^i \right\} = Q_{s-1}.$$

Thus, $\{Q_s : s = 1, ..., t\}$ is a $\mathcal{G}_s$-martingale, so

$$\mathbb{E}[\widehat{Z}_t] = \mathbb{E}[Q_t] = \mathbb{E}[Q_1] = \int \omega_1(x_1) f_1(x_1) q_1(x_1) dx_1 = \int \gamma_1(x_1) f_1(x_1) dx_1$$

$$= \int \gamma_2(x_{1:2}) f_2(x_{1:2}) dx_{1:2} = \cdots = \int \gamma_t(x_{1:t}) f_t(x_{1:t}) dx_{1:t} = Z_t.$$

## Footnotes

[1]https://github.com/freli005/smc-pgm-twist

[3]In a practical implementation it is of course more efficient to skip the resampling step when the ESS is above the threshold. However, this interpretation is useful for the sake of analysis, since it means that we do not need to treat the case with ESS-triggered resampling separately.