[Reviews · NeurIPS 2018]

Reviewer 1



This is a very well-written paper that is beautifully presented with a high degree of clarity. The paper considers a sequential Monte Carlo (SMC) sampler for probabilistic graphical models (PGM). The general methodology is very similar to what was presented by Naesseth et al. (2014) in a previous NIPS conference. The main novelty of the paper lies in the modification of the SMC sampler with a "twisting function." This builds upon recent work in the SMC literature where twisted SMC has recently been considered. The authors propose three possible twisting functions and show the empirical improvement in using twisted SMC on three examples. An important requirement for importance sampling is that the proposal should be heavier tailed than the target form which one wishes to sample. This is necessary to ensure that the importance weights are finite. It wasn't clear from paper how this property is preserved by the twisting proposal. The simulation study is reasonably thorough. However, it is odd that the three twisting functions are not compared against one another. Therefore, it's difficult to get any intuition as to which twisting approach should be used in practice. Overall, this is a very nicely written paper. However, the paper lacks originality and the novelty in the contribution is relatively incremental.

Reviewer 2



Summary and Assessment: ------------------------ This paper strives to improve Sequential Monte Carlo (SMC) sampling on probabilistic graphical models through the usage of twisted targets. More specifically, rather employ a "myopic" sequence of target distributions consisting of gradually introducing the factors and variables in the overall target (according to some ordering criteria) a method is devised by which the future can be conditionally approximated and taken into account. The idea is to devise a target that more closely approximates the true marginal distribution (\pi( x_1,...x_t) of \pi at step t rather than that resulting from dropping all future interactions. Proposition 1 presents the ideal but infeasible choice of twisting function. In effect, equation (6) defines a conditional partition function, and so approximating it with a deterministic method seems sensible. The authors present loopy BP, EP, and Laplace approximation approaches to achieve this. Improvements over myopic SMC are demonstrated in several regimes. This is a good paper with nice ideas; I am encouraged to see ongoing research into improved sampling for PGMs. The paper is very well-written and clearly motivates the problem and contributions. Inference in PGMs remains an important problem today, and the authors present a natural and intuitively appealing enhancement to sampling techniques on these models. I believe this work should be accepted to NIPS. The results and statements are technically correct to my knowledge. Questions and Minor Issues: -------------------------- - Can more be said about the computational burden of the variational fitting? On line 232, the cost of BP is described as "quite small" which may well be true for 16x16 lattices, but the issue is how does such effort scale with problem size. Nonetheless I can readily believe that in many situations, the extra cost will be more than compensated for in the overall efficiency compared to myopic SMC. - This is my most "serious" concern: is there any reason the authors did not include comparisons to SMC under traditional temperature-based annealing? In such a scheme, of course, the overall distribution is always considered, though at different "smoothness" due to the introduction of high-temperature target distributions. It is my experience that for "complex" energy landscapes, it is often quite difficult to do better than a temperature-based sequence. Quite likely, as the authors' results suggest, the cases presently considered are not such "rough" problems, as hinted at by the success of the variational approximations, but in those cases thermal annealing also tends to work quite well. I do suggest that in future work, the authors investigate this if they have not already done so. - A suggestion regarding related work: a relatively early application of SMC to PGMs was Hamze and de Freitas (NIPS '05), where edges (potentials) were sequentially introduced to form the target sequence. - A natural question one may ask, given that the method does something along the lines of a "lookahead", and that equation (6) specifies a conditional partition function, is how well a recursive application of the naive SMC method to approximate the optimal twisted target would fare. More specifically, (6) could be approximated with the myopic SMC (at step t, over variables t+1 to T), but with simulation parameters setting up a fast "pilot" run to get a rough estimate; it would be interesting to see how this does against the variational approximations. Have the authors considered such an approach?

Reviewer 3



The paper describes an approach to make inference over the variables in a probabilistic graphical model by leveraging sequential Monte Carlo method by twisting the target distribution. Twisting of the target distribution allows the future observations to be of influence when making inference, which addresses a key weakness of existing SMC for PGM algorithm (Naesseth et. al. (2014)). The twisted target can be specified using existing inexact, yet fast inference methods for PGM. The authors propose three specific choices that can be useful for specifying the twisting function and they seem to cover wide range of models as demonstrated in the experiments. I think the paper is clearly written and it is easy to follow. But the paper may be slightly lacking in terms of novelty expected from a NIPS paper. Twisting the function is not a new idea, adapting it for PGM and working out the details is not exactly trivial but its contribution appears to be incremental compared to previous work in this field by Naesseth et al. (2014).